# Depth-averaged instantaneous currents in a tidally dominated shelf sea from glider observations

Lucas Merckelbach[1]

[1]Helmholtz Zentrum Geesthacht, Centre for Coastal Research, Geesthacht, Germany.

*Correspondence to:* Lucas Merckelbach (lucas.merckelbach@hzg.de)

**Abstract.** Ocean gliders have become ubiquitous observation platforms in the ocean in recent years. They are also increasingly used in coastal environments. The coastal observatory system COSYNA has pioneered the use of gliders in the North Sea, a shallow tidally energetic shelf sea.

For operational reasons, the gliders operated in the North Sea are programmed to resurface every 3-5 hours. The glider's deadreckoning algorithm yields depth averaged currents, and averaged in time over each subsurface interval. Under operational conditions these averaged currents are a poor approximation of the instantaneous tidal current.

In this work an algorithm is developed that estimates the instantaneous current (tidal and residual) from glider observations only. The algorithm uses a first-order Butterworth low-pass filter to estimate the residual current component, and a Kalman filter based on the linear shallow water equations for the tidal component. A comparison of data from a glider experiment with current data from an ADCP deployed nearby shows that the standard deviations for the east and north current components are better than $7~\mathrm{cm \cdot s^{-1}}$ in near-real time mode, and improve to better than $6~\mathrm{cm \cdot s^{-1}}$ in delayed mode, where the filters can be run forward and backward.

In the near-real time mode the algorithm provides estimates of the currents that the glider is expected to encounter during its next few dives. Combined with a behavioural and dynamic model of the glider, this yields predicted trajectories, the information of which is incorporated in warning messages issued to ships by the (German) authorities. In delayed mode the algorithm produces useful estimates of the depth averaged currents, which can be used in (process-based) analyses in case no other source of measured current information is available.

## 1 Introduction

Ocean gliders, or gliders for short, have become ubiquitous observation platforms in the ocean in recent years. In the Coastal Observing SYstem for Northern and Arctic seas observatory system, COSYNA, (Baschek et al., 2016) the use of Teledyne Webb Research Slocum electric gliders has been pioneered in the North Sea, a tidally energetic shelf sea. The gliders operated within COSYNA are equipped with CTD, optical backscatter, fluorescence and microstructure sensors, intended to observe more or less directly parameters such as temperature, salinity, (proxies for) suspended sediment (via optical backscatter) and chlorophyll $a$ concentrations (via fluorescence), and turbulence dissipation rates.

Gliders have found application in a wide range of research topics, see (Rudnick, 2016) for a recent review. In particular the availability of small size and low power optical backscatter and fluorescence sensors makes gliders suitable for studies on biochemical processes. For example, the occurrence of phytoplankton blooms is strongly influenced by mixing, as mixing affects light conditions and nutrient budgets in the water column.(e.g., Xu et al., 2013). The structure of the water column in shelf seas such as the North Sea, is a balance between stabilising surface heating, and destabilising turbulence generated by shear close to the sea bed and at the surface by tidal currents and wind driven currents, respectively (Simpson and Hunter, 1974). In addition, pelagic mixing caused by shear across the thermocline couples the euphotic and the eutrophic zones above and below the thermocline, and therefore has an important effect on the benthic and pelagic foodweb (Rippeth et al., 2005). Consequently, the analysis and interpretation of the biochemical parameters measured by the glider, requires understanding of the mixing, which in shelf seas, in turn depends strongly on tidal and wind driven currents.

Shelf seas are often shallow enough that tidal and wind driven currents lead to resuspension and deposition events of sediment (Glenn et al., 2008; Tropp, 2013). Similar to fluorescence sensors, optical backscatter or turbidity sensors are commonly fitted to gliders, often even in the same housing. Optical backscatter intensity, when calibrated against filtrated water samples, is a proxy for sediment concentration. Also here, the analysis and interpretation of resuspension events and transport of suspended sediments requires information on the local currents.

Although currents can be measured from gliders using low power acoustic Doppler current profilers (ADCPs) (Johnston et al., 2013), their high cost and (still) relatively high power consumption makes the use of ADCPs on gliders prohibitive, except for dedicated experiments. Instead, moored or shipborn ADCPs could be used, however, this would restrict the moving space of the glider, or the currents measured would not co-locate with the glider data. Alternatively current estimates from the glider itself, resulting from its navigation algorithm could be used (Section 2). However, the loss of information due to averaging in time can become substantial in situations when subsurface times become of the same order of magnitude as the time scale of the variability of the current. As an example, in the North Sea, which is dominated by the semidiurnal M2 tide, the current reverses every 6 hours or so. During glider operations the typical subsurface time is about 3 hours, and therefore poorly resolves the tidal variability.

From an operational point of view, any significant variation in the currents that, from a glider's perspective, appears to have a time scale that is similar to its subsurface time, will cause it to have trouble maintaining the pre-programmed course. Methods have been developed in order to plan trajectories for optimal sampling purposes (e.g., Garau et al., 2009), or to reconstruct the underwater trajectory to localise the data that are gathered by the glider (e.g., Smith et al., 2010). The source of information on the water motion is usually an ocean current model. Elaborating on this work, Smith et al. (2012) developed a system aiming at the effective execution of a planned path, that is, designing the mission such that the glider is capable of travelling along the planned path within given constraints. Effective execution also improves safety at sea, as smaller regions can be defined where gliders can be present.

Safety at sea is in fact a major aspect in glider operations with the COSYNA coastal observatory. Since glider operations take place mostly in the German sector of the North Sea, the planning and execution of glider missions need to comply with the regulations set by the governing German shipping authority Wasser- und Schifffahrtsamt (WSA). This involves the application

for permission to run gliders in a given region within a certain time frame. Since there is a risk of a ship-glider collision (Merckelbach, 2013), which may damage vulnerable fast off-shore vessels (Drücker et al., 2015), WSA requires mitigating measures to be taken by providing the German Vessel Traffic Control Centre (Seewarndienst) with 12-hourly forecasts of the region where the glider will be, given by the four coordinates defining a rectangle. The system that has been set up to provide these forecasts (not discussed herein) relies on a model simulating the behaviour of the glider by emulating the glider software and hardware, as well as modelling the dynamics (*i.e.,* its flight through water). However, a realistic prediction requires information on the local currents up to 12 hours ahead.

In the present work ocean current models are not relied upon to provide information on the water motion. Instead, the aim is to reconstruct the instantaneous currents by recovering (most of) the information in the observed currents lost due to the time averaging.

To that end, an algorithm is proposed that is composed of a simple low pass filter for low frequency variations in the currents due to atmospheric influence, for example, and a Kalman filter based on the shallow water equations to estimate the tidally induced variation in the currents.

## 2 Depth and time averaged currents from the glider platform

The Teledyne Webb Research Slocum electric glider uses a dead-reckoning algorithm for underwater positioning. The algorithm combines the depth rate of change from the pressure transducer and heading and pitch from the attitude sensor to compute the horizontal velocity components. The dead-reckoned underwater position follows from integrating the current vectors with respect to time, starting from the latest known GPS position. The difference between the dead-reckoned resurface position and the actual GPS position is attributed to a depth and time averaged current, see also Merckelbach et al. (2008), for example. The glider user can define whether or not the glider navigation algorithm should apply this current estimate to compensate for drift when calculating the heading for the current waypoint during the next dive. Doing so only makes sense, however, when the time variability is sufficiently resolved.

## 3 An algorithm for short-time current estimates

In this work, estimates of time averaged currents are used to reconstruct instantaneous currents. The reconstructed instantaneous time series contain more information than the observed time averaged currents. Since the extra information required does not come from other measurements or observations, it will have to be provided by a model.

The currents in a coastal sea as considered herein, are dominated by the tide, so that a (simple) model, such as the shallow water equations, can provide this additional information on the tidal motion. Besides tidal currents, non-tidal currents due to atmospheric conditions and fresh water influx, for example, can have significant effects. The non-tidal or residual current on the other hand, is hard to model without resorting to complex numerical models. Therefore the current is decomposed in a tidal current and a slowly varying residual current. The tidal current component is then estimated from a Kalman filter based

on the shallow water equations, including at most a few of the main tidal constituents. The slowly varying residual current is estimated by removing the semidiurnal tidal components and their higher harmonics using a low pass filter.

## 3.1 Residual currents

Generally, low pass filters show a gradual the transition from pass to no-pass at the cut-off frequency (the transition band). In addition, low pass filters introduce a frequency dependent lag to the filtered signal (the group delay) so that the phase of the signal is not preserved. The design of a low pass filters is a trade-off between how broad the transition band is allowed to be, and how much lag is acceptable.

For this purpose, an efficient low pass Butterworth filter is implemented (e.g., Oppenheim et al., 1997). The properties of the filter are determined by cut-off frequency $f_c$ and its order $N$. The order of the filter determines the width of the transition band and affects the group delay. Figure 1 shows the filter responses of a number of Butterworth filters of the order $N = \{1,2\}$. The top panel shows the (power) gain as a function of frequency and the bottom panel the group phase delay as function of frequency.

In order to effectively remove the main tidal signals (semidiurnal components and their higher harmonics), the cut-off frequency of each filter can be chosen such that the gain of frequencies with a period of 12 hours or smaller have a gain less than 0.01, yielding cut-off frequencies of $f_c = \{1/119, 1/38\}$ cph for $N = \{1,2\}$. The figure shows that a higher order filter has a narrower transition band, but also has a larger (detrimental) effect on the group delay. The effects of increasing the cut-off frequency is that less of the tidal signal is damped, but also that the group delay is reduced, as is shown in the figure for the cut-off frequencies $f_c = 1/24$ and $f_c = 1/12$ cph.

In a practical application the filter can be implemented such that every time a new current measurement becomes available, that is, when the glider resurfaces, the measured value is fed into the low pass filter, and yields an estimate of the residual current. This estimate has an error because of a time delay, but also because of tidal signals that may still be present if the attenuation is insufficient. In the post-processing, when all data are available, the filtering can be improved by running the filter forward and backward: the time delay introduced is compensated, and unwanted signals are further damped.

At this stage it is not clear what filter setting would give the optimum results, so that, for now, a low pass filter is designed with $N = 1$ and $f_c = 1/24$ cph. Although this filter passes about 40% of the semidiurnal tidal signal, corresponding to a power gain of about 0.2, the filter setting introduces little group delay, see also Figure 1. In Section 6 the implications of this choice are discussed.

## 3.2 Tidal currents

In contrast to the residual currents, the evolution of tidal currents can be captured to a large extent by a simple model. In this section we will cast such a simple model into a Kalman filter to provide an optimal estimate of the tidal current components that the glider will face during its next dive, based on all previous depth and time-averaged water current data it has collected.

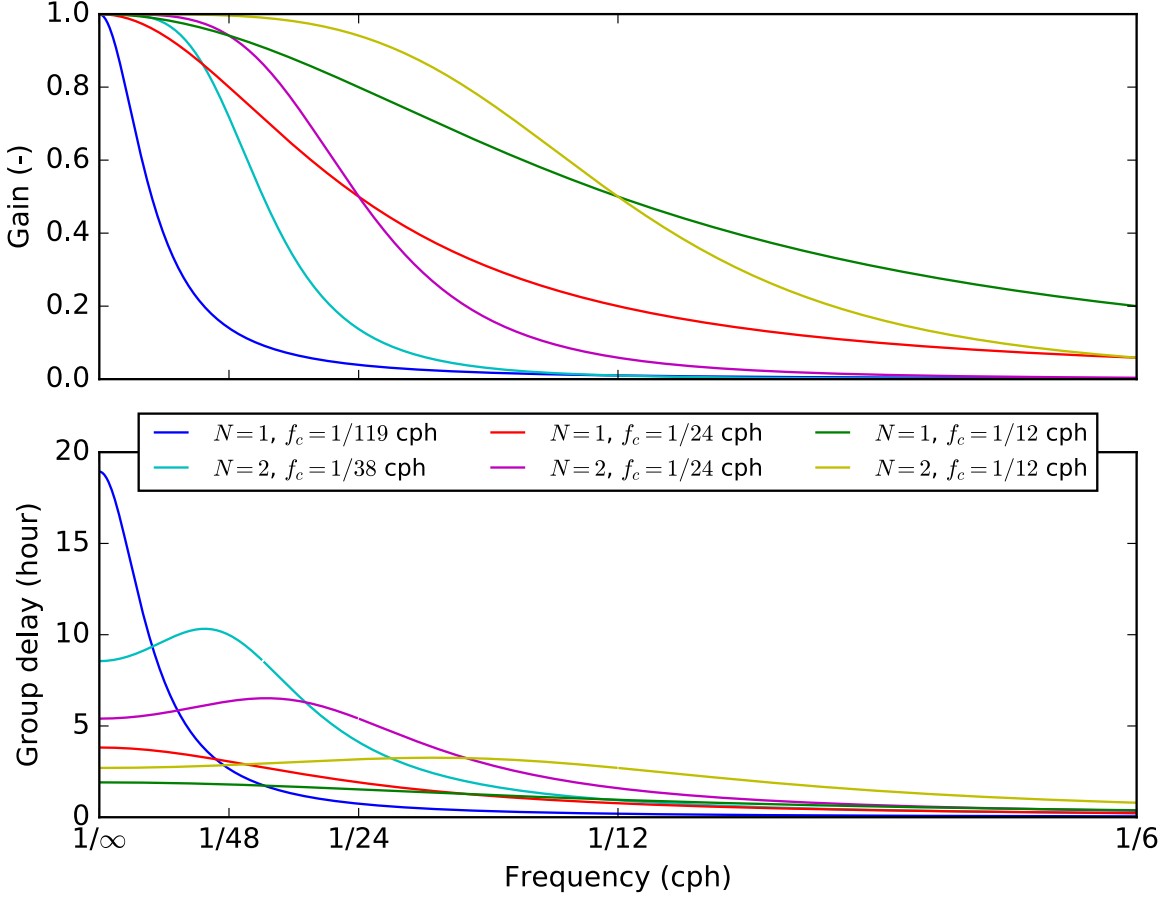

**Figure 1.** Filter responses of Butterworth filters of the first and second order for various cut-off frequencies. Top panel shows the power gain as a function of frequency, and the bottom panel shows the group delay as a function of frequency.

### 3.2.1 Intermezzo on Kalman filtering

For an introduction on Kalman filters and their derivation, the reader is referred to e.g. Simon (2006). Here, for the sake of brevity, we will state the general computational procedure only.

The Kalman filter is formulated as a dynamical system (Simon, 2006)

$$\boldsymbol{x}_k = \boldsymbol{F}_{k-1}\boldsymbol{x}_{k-1} + \boldsymbol{w}_{k-1}$$
$$\boldsymbol{y}_k = \boldsymbol{H}_k\boldsymbol{x}_k + \boldsymbol{v}_k$$
$$\boldsymbol{w}_k \approx \mathcal{N}\{0, \boldsymbol{Q}_k\}$$
$$\boldsymbol{v}_k \approx \mathcal{N}\{0, \boldsymbol{R}_k\} \tag{1}$$

where $\boldsymbol{x}$ is the state vector, $\boldsymbol{F}$ the transition matrix, $\boldsymbol{y}$ the measurement vector, $\boldsymbol{H}$ the measurement matrix, $\boldsymbol{w}$ the process noise vector (normal distributed with zero mean and known variance $\boldsymbol{Q}_k$), $\boldsymbol{v}$ the measurement noise vector (normal distributed with zero mean and known variance $\boldsymbol{R}_k$) and $k$ the measurement index number. If, for example, the state vector is composed of the eastward and northward velocity components, then the transition matrix describes how the currents would change from one time step $k$ to the next. The measurement matrix relates the observed parameters, $\boldsymbol{y}$, to the state vector.

The procedure for the Kalman filter is given by the following equations for $k = 1, 2, 3, ...$

$$\boldsymbol{P}_k^- = \boldsymbol{F}_{k-1}\boldsymbol{P}_{k-1}^+\boldsymbol{F}_{k-1}^T + \boldsymbol{Q}_{k-1} \tag{2}$$
$$\boldsymbol{K}_k = \boldsymbol{P}_k^-\boldsymbol{H}_k^T(\boldsymbol{H}_k\boldsymbol{P}_k^-\boldsymbol{H}_k^T + \boldsymbol{R}_k)^{-1} \tag{3}$$
$$\hat{\boldsymbol{x}}_k^- = \boldsymbol{F}_{k-1}\hat{\boldsymbol{x}}_{k-1}^+ \tag{4}$$
$$\hat{\boldsymbol{x}}_k^+ = \hat{\boldsymbol{x}}_k^- + \boldsymbol{K}_k(\boldsymbol{y}_k - \boldsymbol{H}_k\hat{\boldsymbol{x}}_k^-) \tag{5}$$
$$\boldsymbol{P}_k^+ = (\boldsymbol{I} - \boldsymbol{K}_k\boldsymbol{H}_k)\boldsymbol{P}_k^-(\boldsymbol{I} - \boldsymbol{K}_k\boldsymbol{H}_k)^T + \boldsymbol{K}_k\boldsymbol{R}_k\boldsymbol{K}_k^T, \tag{6}$$

where $\hat{\boldsymbol{x}}_k^-$ is the *a priori* estimate, $\boldsymbol{P}_k^-$ the *a priori* covariance, $\hat{\boldsymbol{x}}_k^+$ the *a posteriori* estimate, $\boldsymbol{P}_k^+$ the *a posteriori* covariance, $\boldsymbol{K}_k$ the gain matrix, and $\boldsymbol{I}$ the identity matrix.

The procedure involves sequentially evaluating the set of equations (2) – (6) every time a measurement vector becomes available. First the measurement index $k$ is advanced by 1. Then the *a priori* covariance estimate is computed from the *a postiori* covariance estimate and the transition matrix at the previous index, (2). Subsequently, the *a priori* estimate of the state vector is computed from the transition matrix and the *a postiori* estimate of the state vector at the previous level, (4), that is, only the model is used to produce a first estimate of the new state vector. The estimate of the state vector is further improved (to yield the *a postiori* estimate) by including the measurement vector. The improvement is equal to the difference of the measured value and the modelled value (*a priori* estimate), multiplied by the gain matrix $\boldsymbol{K}_k$, (5). Herein the gain matrix is computed previously in the second step (3). In the final step the estimate of the covariance is improved (the *a postiori* covariance estimate) using (6). The cycle is repeated with the arrival of the next measurement vector.

The filter is initialised with estimates for

$$\hat{\boldsymbol{x}}_0^- = \hat{\boldsymbol{x}}_0, \text{ and} \tag{7}$$

$$\boldsymbol{P}_0^- = \boldsymbol{P}_0. \tag{8}$$

If no *a priori* information on the system is available, the state vector can be set to zeros, accompanied by a relatively high
valued diagonal covariance matrix, expressing the uncertainty of the initial guess.

### 3.2.2  Kalman filter formulation

For a model to capture the main tidal oscillation, it is assumed that the shallow water equations are a reasonable model for the
water dynamics:

$$\begin{aligned}
\frac{\partial u}{\partial t} - fv + g\frac{\partial \eta}{\partial x} &= 0 \\
\frac{\partial v}{\partial t} + fv + g\frac{\partial \eta}{\partial y} &= 0,
\end{aligned} \tag{9}$$

where $u$ and $v$ are the eastward and northward velocity components, respectively, $x$ and $y$ the eastward and northward coordinates, respectively, $f$ the Coriolis parameter, $\eta$ the surface elevation and $g$ the acceleration due to gravity. The shallow water
equations (9) can be cast as

$$\begin{aligned}
\frac{\partial^2 u}{\partial t^2} + f^2 u &= -g\left(f\frac{\partial \eta}{\partial y} + \frac{\partial^2 \eta}{\partial t\partial x}\right) \\
\frac{\partial^2 v}{\partial t^2} + f^2 v &= -g\left(f\frac{\partial \eta}{\partial x} + \frac{\partial^2 \eta}{\partial t\partial y}\right)
\end{aligned} \tag{10}$$

The right-hand side terms in (10) can be regarded as the forcing of the system, which we try to seek. To that end, the surface
level gradients are represented by harmonic functions with unknown coefficients $A_{\{x,y\}}$ and $B_{\{x,y\}}$,

$$\frac{\partial \eta}{\partial x} = A_x \cos\omega t + B_x \sin\omega t, \text{ and} \tag{11}$$

$$\frac{\partial \eta}{\partial y} = A_y \cos\omega t + B_y \sin\omega t, \tag{12}$$

where $\omega$ is the main tidal frequency ($M_2$, for example). The addition of more tidal frequencies is trivial, however.

Defining the dynamical system (1) the state vector is chosen as

$$\boldsymbol{x} = [A_x, B_x, A_y, B_y]^T. \tag{13}$$

Furthermore, the coefficients $A_{\{x,y\}}$ and $B_{\{x,y\}}$ are modelled as constants, so that, as an example, the process for $A_x$ becomes

$$A_{x\,k} = A_{x\,k-1} + \text{ process noise}, \tag{14}$$

leading to the simple transition matrix

$$\boldsymbol{F} = \mathrm{diag}(1,1,1,1). \tag{15}$$

The measurements ($\boldsymbol{y}$) are depth averaged water velocities, averaged in time from the time of diving until the time of resurfacing. In order to determine the elements in the measurement matrix $\boldsymbol{H}$ that relates the measurements to the state vector as $\boldsymbol{y} = \boldsymbol{Hx}$, the surface elevation gradients (11) and (12) are substituted into (10), which gives the expressions for the instantaneous current components

$$u = \frac{1}{f^2 - \omega^2} \left[ a_0 \cos \omega t + a_1 \sin \omega t \right] \tag{16}$$

$$v = \frac{1}{f^2 - \omega^2} \left[ b_0 \cos \omega t + b_1 \sin \omega t \right] \tag{17}$$

where

$$a_0 = -gf A_y - g\omega B_x$$

$$a_1 = -gf B_y + g\omega A_x$$

$$b_0 = -gf A_x - g\omega B_y$$

$$b_1 = -gf B_x + g\omega A_y$$

The averaged currents are then found by integrating the instant current components with respect to time and dividing the result by the subsurface time $T$. It follows that

$$\boldsymbol{H} = \frac{g}{T(f^2 - \omega^2)} \begin{bmatrix} -C & -S & -f/\omega S & +f/\omega C \\ -f/\omega S & +f/\omega C & -C & -S \end{bmatrix}. \tag{18}$$

Herein $C = \cos(\omega t_1) - \cos(\omega t_0)$ and $S = \sin(\omega t_1) - \sin(\omega t_0)$, in which $t_0$ and $t_1$ are the dive and resurface times, respectively, and $T = t_1 - t_0$.

The measurement error is assumed to be directionally uncorrelated, so that

$$\boldsymbol{R} = r\mathrm{diag}(1,1), \tag{19}$$

where $r$ is the variance of the measurements. The numerical value can be estimated relatively easy. Factors that influence the accuracy of the measurement are the accuracy of the dive and resurface positions, and the dead-reckoning algorithm. Dive and resurface positions are derived from GPS measurements, which have a finite precision, roughly 10-20 m. However, since it takes time for a valid GPS position to be received, the resurface position and the position when the first valid GPS value is acquired do not necessarily co-locate. In addition, the dead-reckoning algorithm uses input data, such as depth rate, heading and pitch, provided by several sensors, each introducing a degree of uncertainty. A value of $1 \ \mathrm{cm \cdot s^{-1}}$ is taken as a reasonable estimate for the accuracy of the depth and time averaged current measurement from the glider platform.

It is less obvious how to quantify the process noise matrix $\boldsymbol{Q}$. The process noise accounts for uncertainties in the model description of the process. Clearly, using the steady state solution of the shallow water equations, assuming the only forcing is due to the tides, the model underpinning the Kalman filter is not fully representing reality. The process noise allows for some distrust in the model description, favouring the measurements, or, put differently, it allows for flexibility of the model to adapt by "forgetting" old measurements. It is assumed that the process noises for each tidal component and direction is uncorrelated, so that

$$\boldsymbol{Q} = q\text{diag}(1,1,1,1). \tag{20}$$

The variance parameter $q$ is then regarded as a tuning parameter. The value is considered optimal when the variance or standard deviation of $\varepsilon$ is minimal. The elements in $\varepsilon$ are given by (see also (5))

$$\varepsilon_k = \boldsymbol{y}_k - \boldsymbol{H}_k \hat{\boldsymbol{x}}_k^-, \tag{21}$$

that is, the averaged current estimate is computed from the measurement matrix $\boldsymbol{H}$ at the current level, but with the estimate for the amplitudes of the previous level/surfacing. Note that due to the virtue of (15), $\hat{\boldsymbol{x}}_k^- = \hat{\boldsymbol{x}}_{k-1}^+$.

## 4   Assessment of the performance of the algorithm

### 4.1   Instrumentation and field data

Below, data of measured currents are used to assess the performance of the current prediction algorithm. Two data sets were used for this purpose. The first data set was obtained from measured currents from a bottom mounted acoustic Doppler current profiler, from which synthetic, but realistic, depth and time averaged currents were constructed. These data time series mimick the glider data, but have predefined and controllable subsurface times. In addition, the synthetic data set removes any uncertainty introduced by the glider's dead-reckoning algorithm, rendering this a useful data set to assess the performance of the prediction algorithm *per se*. The second data set uses current estimates from glider data obtained during a field experiment. Analysing the results for both data sets allows us to quantify the effects on accuracy of the subsurface time and the glider's dead-reckoning algorithm.

The data used in this study were collected during a field experiment that took place in the German Bight, in the German sector of the North Sea in August 2014, see Figure 2a. An upwardlooking RDI Workhorse 600 kHz ADCP was bottom mounted near buoy NSB3 ($54°40.7'$ N, $6°47.1'$ E) at about 40 m depth (Figure 2a). The deployment period ranges from 16 July 2014 until 31 August 2014. The instrument was configured with bin sizes of 40 cm yielding a current profile every 10 minutes. Each measured profile is the ensemble mean of 32 pings.

The glider *Sebastian*, a Teledyne Webb Research Slocum Electric littoral glider (Jones. et al., 2005), was deployed on 24 July 2014 and recovered on 26 August 2014. Its track is shown in Figure 2a. From 4 August until 10 August and from 13 August until 17 August, the glider was programmed to fly in a spiralling mode (with the steering fin set to a fixed position).

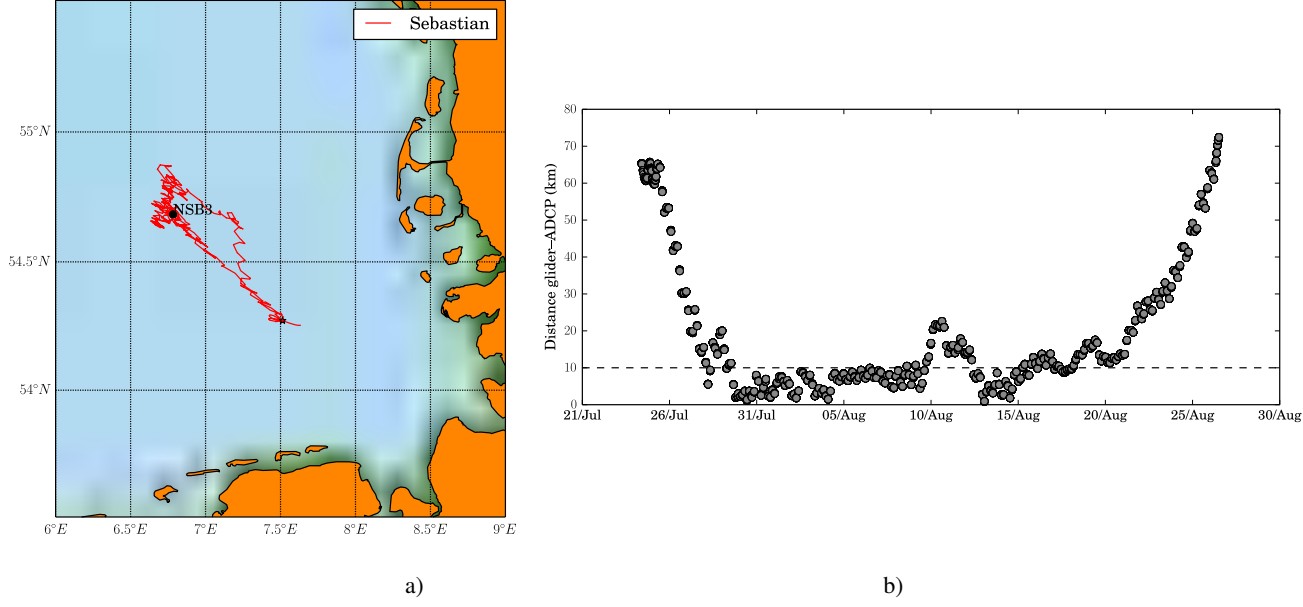

a)                                                    b)

**Figure 2.** Panel a) Track of glider *Sebastian* during 2014 experiment and location of the bottom mounted ADCP near buoy NSB3; panel b) distance between glider and bottom mounted ADCP.

The target area for the glider operation was near the buoy NSB3 and the ADCP. Figure 2b shows that most of the time the glider was within 10 km distance from the ADCP. Therefore, the tidal currents and the mesoscale circulation that are subject to the ADCP and glider measurements are expected to be the same.

### 4.2  Algorithm assessment

As a first step, the ADCP measurements were used to evaluate the performance of the filter. The instantaneous currents measured with the ADCP are considered as the true currents. Synthetic glider measurement data were obtained by time and depth averaging the ADCP measurements, followed by adding white noise. The time averaging was performed over a window representing the subsurface time of the glider. This interval was set to 3 hours, which is a typical value during operations in the North Sea. Below, however, the influence of the subsurface time on the accuracy on the prediction algorithm is addressed specifically.

The added noise is Gaussian with zero mean and a standard deviation of $1\ \mathrm{cm}\cdot\mathrm{s}^{-1}$, in correspondence with (19).

Firstly, the synthetic measurements were low pass filtered using a first order Butterworth filter as outlined above. The purpose of this filter is to remove the main semidiurnal and faster tidal signatures from the total signal. For the present data set the M2 tidal component accounts for 80% and 65% of the total variance for the eastward and northward currents, respectively. The result is shown in Figure 3 for the forward filter (red) and the phase-preserving forward-backward filter (green). The synthetic

measurement data are shown in transparent blue. The figure shows that the residual current does not vary too much, except for

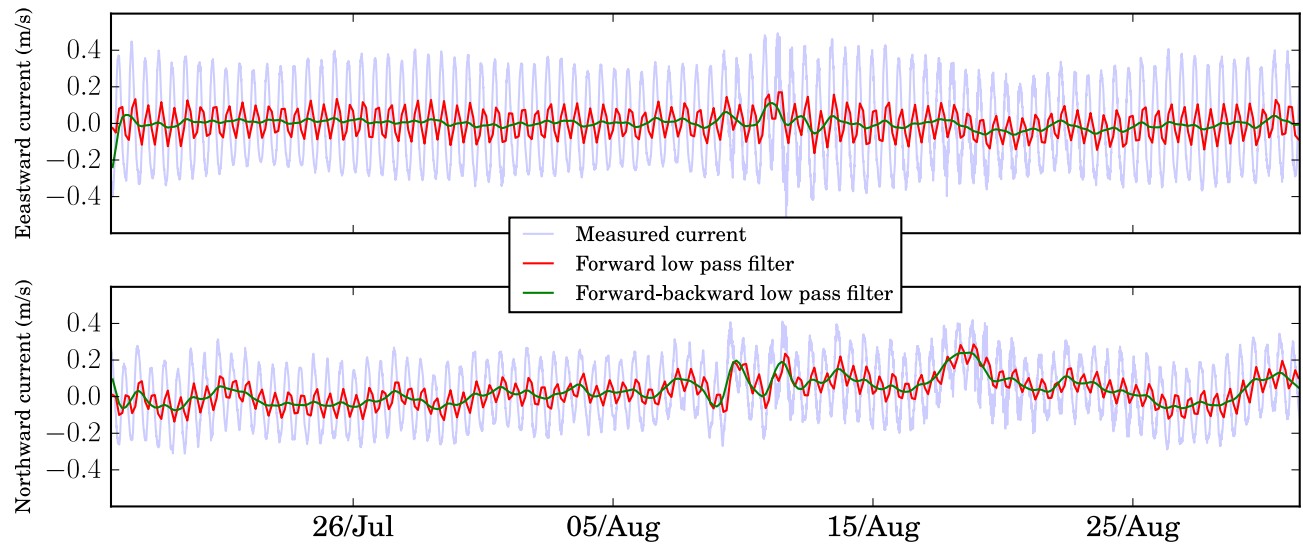

**Figure 3.** Eastward and northward currents are shown in the top and bottom panel, respectively. The synthetic 3-hour averaged currents are shown in transparent blue, and the forward and forward-backward filtered residual currents are shown in red and green, respectively.

a few short periods, notably in the northward current. One of these periods with significant variation in the residual current is the time window of some 3 days starting at 10 August, which can be linked to the passage of a low pressure system (remnants of the hurricane "Bertha"). The graphs show that, compared to the forward-backward filter, the forward filter leaves about 40% of the main semidiurnal tidal component present in the filtered signal, but introduces only a marginal phase lag.

Secondly, the synthetic current measurements, corrected for the residual current using the low pass filtered currents, were subjected to the Kalman filter. The initial conditions were set by the state vector $\boldsymbol{x}_0 = [0,0,0,0]^T$ (zero tidal amplitude components) and a high covariance matrix $\boldsymbol{P}_0 = \mathrm{diag}(1000, 1000, 1000, 1000)\ \mathrm{m}^2 \cdot \mathrm{s}^{-2}$, which signifies that the current state of the system is unknown. Furthermore, the measurement noise variance parameter was set to $r = 1 \times 10^{-4}\ \mathrm{m}^2 \cdot \mathrm{s}^{-2}$, in correspondence to the added noise, $r = \sigma^2$.

In order to find the optimal value for the variance parameter $q$, the Kalman filter was run for a number of different values of $q$. The Kalman filter yielded the lowest standard deviation of the error in the estimated currents in both near-real time mode and delayed mode (see Section 4.3) for $q \approx 4 \times 10^{-16}$, see Figure 4. The value of $q = 4 \times 10^{-16}$ is used throughout this work.

The Kalman filter (2) – (6) was updated when a new current measurement becomes available, *i.e.* at 3 hour intervals. The error of the estimate of the depth and time averaged current components for each time step is given by (21). The results are

summarised in histograms in Figure 5. The top and bottom left panels show the histograms of the eastward and northward (time integrated) currents, respectively. The distributions of the errors seem Gaussian and have standard deviations of 3.5 and 3.1 $\mathrm{cm} \cdot \mathrm{s}^{-1}$ for the eastward and northward currents, respectively. The means are $\approx 0$ for both the eastward and northward

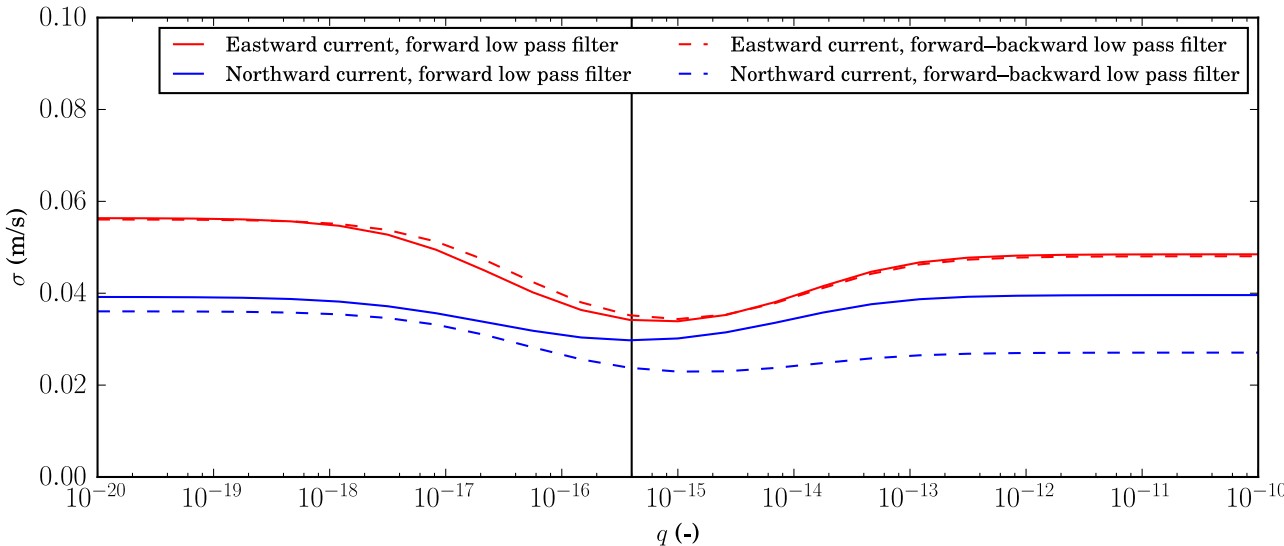

**Figure 4.** Standard deviations of the errors of the eastward (blue) and northward (red) current components, as a function of the variance parameter $q$ (process noise). The solid lines were obtained using a forward low pass filter to remove the residual current (near-real time mode), whereas the dashed lines were obtained using a forward–backward low pass filter (delayed mode). The black line indicates $q = 4 \times 10^{-16}$, which is the optimal value taking into account eastward and northward currents as well as near-real time and delayed mode scenarios.

components. The panels on the right-hand side show the cumulative probability density. The mean error is approximately 2.0 and 2.5 $\mathrm{cm \cdot s^{-1}}$ for the eastward and northward components, respectively. Furthermore, 95% of the estimates have an error margin smaller than 6.9 and 5.8 $\mathrm{cm \cdot s^{-1}}$ for the eastward and northward components, respectively. Put into context this means that for 3 hour subsurface times, about 95% of the estimates of the resurfacing positions are accurate within 1000 m, see also the application of a virtual AIS in Section 5.

In practice, the errors due to the dead-reckoning algorithm that get absorbed into the current measurements by the glider, will degrade the performance of the current prediction algorithm. To quantify this degradation, the algorithm is applied to the observed current measurements from the glider, and compared with the currents as measured by the ADCP, which are considered the ground truth. Since the glider operated within 10 km of the ADCP during most of the mission time, see Figure 2b, this seems a reasonable assumption.

In contrast to the synthetic data, the glider data do not have a fixed interval. For the present glider dataset, most of the subsurface times were between 2.6 and 3.0 hours, as during most of the mission the glider was programmed to resurface at three hour intervals, interpreted as resurface time-to-resurface time. The reason for the mean subsurface time to be less than 3 hours is due to the fact that at resurfacing the glider spent about 10-15 minutes afloat to transmit data. Leaving all parameter settings of the low pass filter and the Kalman filter unchanged, the depth and time averaged current estimates are compared

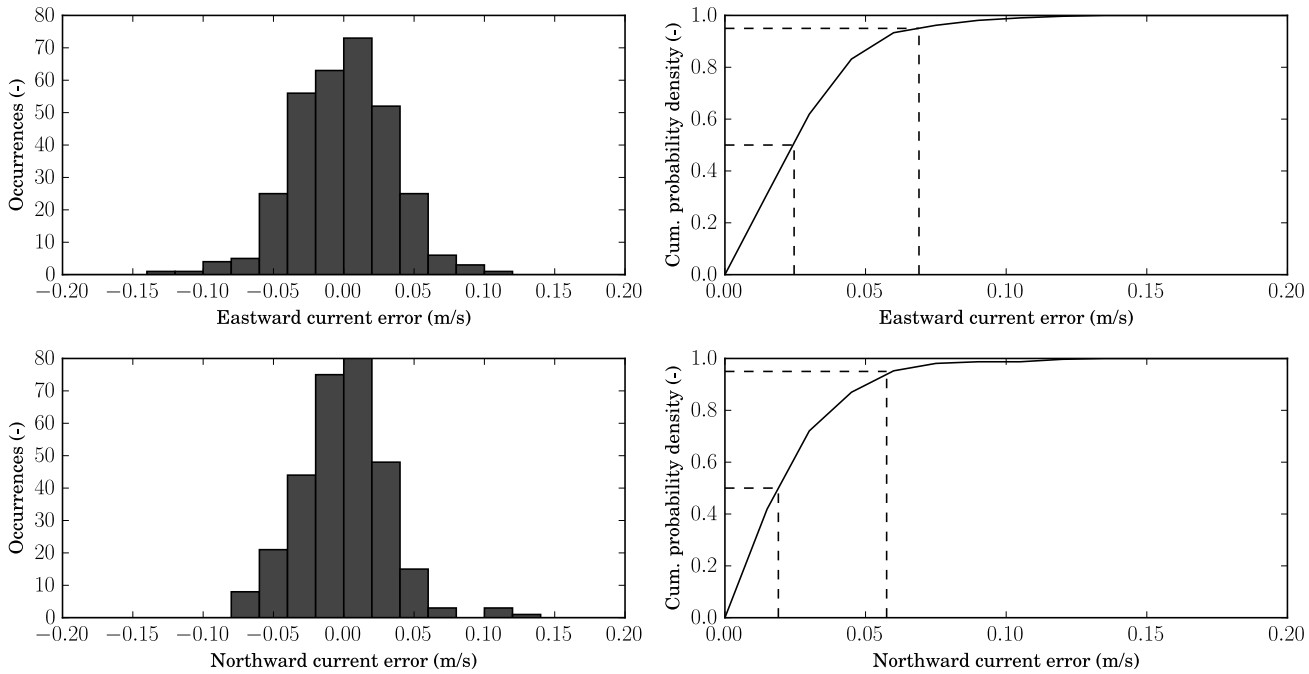

**Figure 5.** Errors in the estimated currents. Upper left panel: histogram for eastward direction, bottom left panel: histogram for northward direction, upper right panel: estimated probability density function for eastward current, and bottom right panel: estimated probability density function for northward current. The dashed lines indicate the 50% and 95% levels.

with those computed from the ADCP current measurements, where time intervals for averaging the ADCP data were matched to the factual subsurface times of the glider. The results are summarised in Figure 7. In this figure the left panel represents the eastward current error and the right panel the northward current error. Comparing the near-real time glider data results (black curves), that is forward-filtered only, with the results of the synthetic dataset from ADCP data (red curves), it is seen that the performance dropped, as expected. On average the errors (50% and 95% levels) increase by a factor of 1.4.

The *instantaneous* currents are readily computed once the amplitude estimates of the tidal components are computed. At surfacing, when a new time and depth averaged current estimate becomes available, the low pass filter and the Kalman filter provide new estimates for the residual current component and the *a postiori* state vector, respectively. The residual current component during the dive is computed from linear interpolation of the estimates obtained prior to diving and just after resurfacing. The time varying tidal component is computed from (16) and (17), with the amplitudes linearly interpolated from the *a postiori* estimates, also obtained just before diving and just after resurfacing. Figure 6 shows an example of a 17-day period of instantaneous currents (synthetic dataset). In particular for the north component, one instance where the residual current

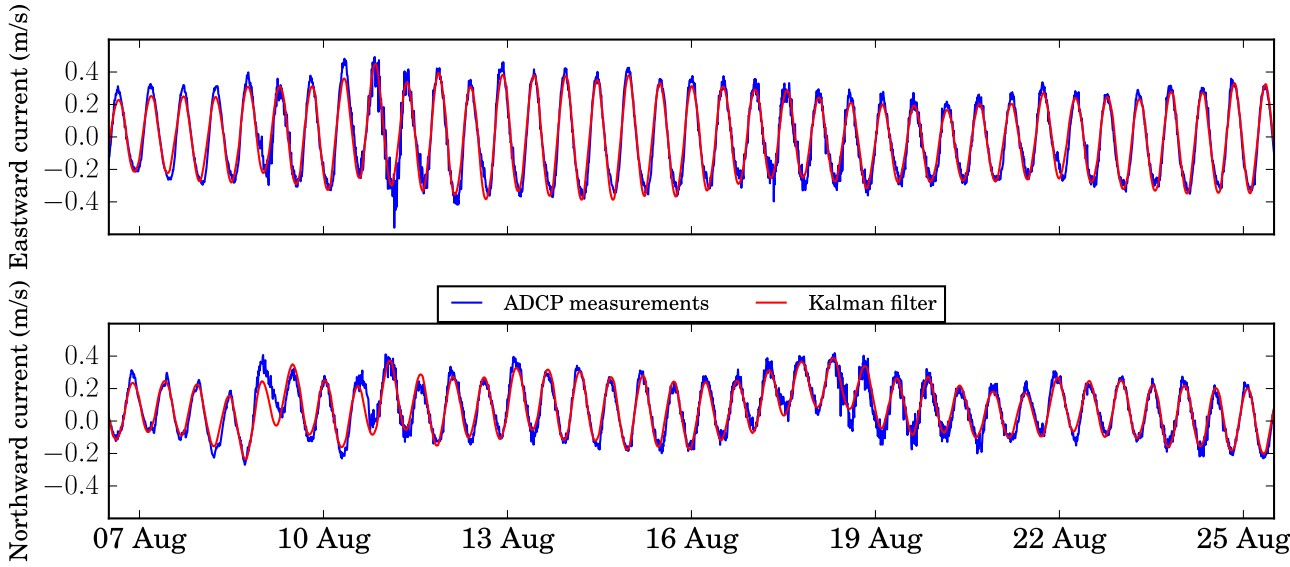

**Figure 6.** 17-Day time series of measured currents and Kalman filter estimated currents for the eastward component (top panel) and the northward panel (bottom panel).

changes in time is discernible, namely around 8 August. Due to the lagging response of the Butterworth filter, the estimated currents deviate most from the measured currents when sudden changes occur in the residual currents.

The near-real time instantaneous current estimates are potentially useful for assimilation into circulation models, see for example Stanev et al. (2015). Every time the glider surfaces, the low pass filter and the Kalman filter can be run, using the
latest available measurement estimate of the depth and time averaged current. In this way, estimates of the instantaneous current during the dive can be used in the assimilation process, whereas depending on the subsurface time, time averaged value of the current may provide little useful information.

Comparing the estimates of the instantaneous (depth-averaged) current components with the instantaneous currents measured with the ADCP, the standard deviations of the differences amount to $6.5 \ \mathrm{cm \cdot s^{-1}}$ for both the eastward and northward
components. The linear correlation coefficients for the observed and estimated current components are in the range $[0.93, 0.97]$, confirming the strong linear relationship between estimates and observations suggested by the data in Figure 6. A summary of mean and standard deviations of the differences between current estimates and observations in near-real time mode is given in Table 1, top panel.

It is noted that, since for the synthetic data set the "measurement data" and "observations" are constructed from the same
source, namely the ADCP currents, any bias in ADCP currents will go unnoticed. Indeed, the mean value of the difference between estimated and observed current for the synthetic data set amounts to 0, see Table 1. For the glider data set on the other

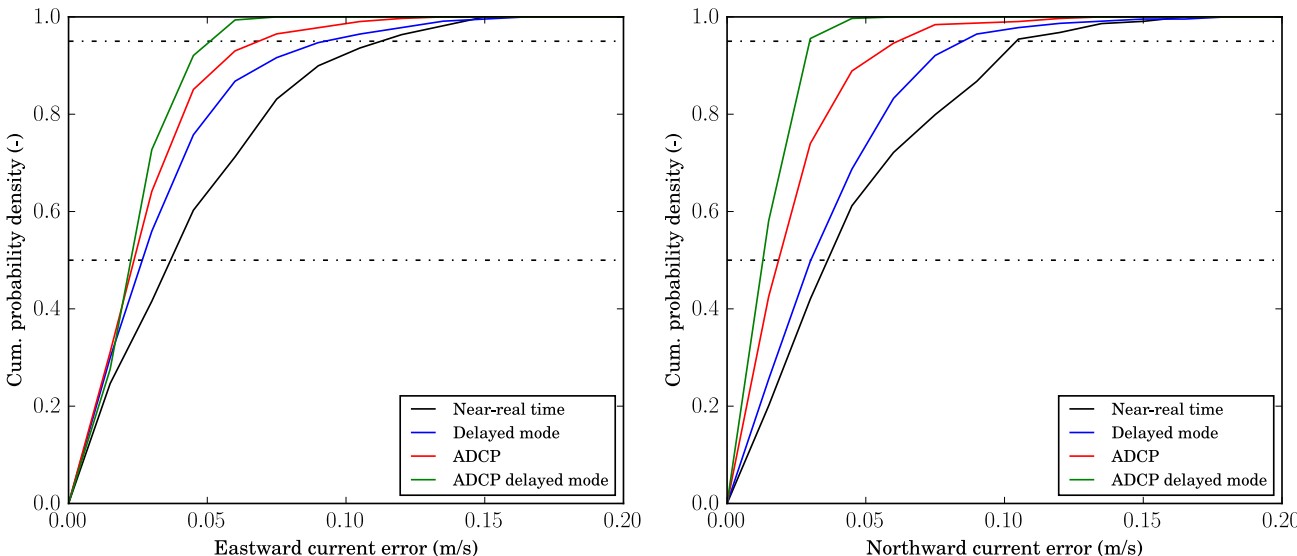

**Figure 7.** Probability density function estimates of the errors for the eastward current component (left panel) and the northward component (right panel). The black and red curves are derived from the results of the Kalman filter run in near-real time mode for the glider data and ADCP data, respectively. The blue and green curves are derived from the Kalman filter modified for delayed mode (post-processing), see Section 4.3.

hand, the table shows that the mean values are not equal to 0. In this case the measurement data (glider) and reference data (ADCP) are independent. The non-zero means can be caused by a bias in the ADCP measurements, which is in the order of 1 $\mathrm{cm \cdot s^{-1}}$, or a bias in the dead-reckoning algorithm of the glider (see also Section 4.3).

### 4.3 Glider derived currents in delayed mode (post-processing)

5    The approach proposed herein can also be used to reprocess the glider data to obtain estimates of the instantaneous barotropic currents once the glider mission has been completed. In delayed mode, a number of improvements can be applied. First, the depth and time averaged current estimates can be improved by recalculating the dead-reckoned position. The glider's dead-reckoning algorithm computes the horizontal velocity component from the pitch and the pressure rate, ignoring the angle of attack. Although the angle of attack is generally small, the glider algorithm may overestimate its horizontal speed by a few 10    $\mathrm{cm \cdot s^{-1}}$. An improved dead-reckoning calculation can be done post-mission by implementing the dynamical glider model of Merckelbach et al. (2010).

Another source of error in the estimated currents is the phase lag introduced by the Butterworth filter, see Section 3.1. This effect can be mitigated by running the filter forwards and backwards in time, as demonstrated in Figure 3.

Near-real time

| Dataset | Time base | Eastward current | | | Northward current | | |
|---|---|---|---|---|---|---|---|
| | | $\mu$ | $\sigma$ | $\rho$ | $\mu$ | $\sigma$ | $\rho$ |
| | | $(\mathrm{cm\cdot s^{-1}})$ | $(\mathrm{cm\cdot s^{-1}})$ | (-) | $(\mathrm{cm\cdot s^{-1}})$ | $(\mathrm{cm\cdot s^{-1}})$ | (-) |
| Synthetic data | Dive averaged | 0.0 | 3.5 | | 0.0 | 3.1 | |
| | Instantaneous | 0.0 | 4.8 | 0.97 | 0.0 | 4.1 | 0.96 |
| Glider data | Dive averaged | 1.4 | 5.2 | | 0.2 | 5.7 | |
| | Instantaneous | 1.5 | 6.5 | 0.96 | 0.8 | 6.5 | 0.93 |

Delayed mode

| Dataset | Time base | Eastward current | | | Northward current | | |
|---|---|---|---|---|---|---|---|
| | | $\mu$ | $\sigma$ | $\rho$ | $\mu$ | $\sigma$ | $\rho$ |
| | | $(\mathrm{cm\cdot s^{-1}})$ | $(\mathrm{cm\cdot s^{-1}})$ | (-) | $(\mathrm{cm\cdot s^{-1}})$ | $(\mathrm{cm\cdot s^{-1}})$ | (-) |
| Synthetic data | Dive averaged | 0.0 | 1.6 | | 0.0 | 1.1 | |
| | Instantaneous | 0.0 | 3.5 | 0.99 | 0.0 | 2.6 | 0.99 |
| Glider data | Dive averaged | 0.7 | 4.3 | | 1.6 | 4.6 | |
| | Instantaneous | 0.6 | 5.7 | 0.97 | 1.6 | 5.5 | 0.95 |

**Table 1.** Mean ($\mu$) and standard deviation ($\sigma$) of the difference between ADCP observations and a) *near-real time* current estimates (top table) and b) *delayed mode* current estimates (bottom table). The correlation coefficient ($\rho$) is calculated from the instantaneous estimated values and observations, for both the near-real time and delayed mode data. The synthetic data set is derived from ADCP measurements, with added noise (see text). Dive averaged current estimates assume 3-hour dives.

Third, a Kalman filter can be formulated that uses both "historic" and "future" observations. To that end the Kalman filter described above is run forward and backward, whereas the final estimate of the vector $\hat{\boldsymbol{x}}_k$ for time index $k$ is combined from the forward and backward quantities (Simon, 2006):

$$\boldsymbol{K} = \boldsymbol{P}_{\mathrm{b},k}^{-}(\boldsymbol{P}_{\mathrm{f},k}^{+} + \boldsymbol{P}_{\mathrm{b},k}^{-})^{-1}, \tag{22}$$

$$\hat{\boldsymbol{x}}_k = \boldsymbol{K}\boldsymbol{x}_{\mathrm{f},k}^{+} + (\boldsymbol{I} - \boldsymbol{K})\boldsymbol{x}_{\mathrm{b},k}^{-}, \tag{23}$$

where the subscripts "f" and "b" denote forward and backward filter results, respectively.

Figure 7 shows the improvement achieved due to additional backward filtering step. For both the synthetic and the glider data sets, the errors in the depth and time averaged currents are reduced. The averaged factor of improvement for the synthetic and glider data is approximately 1.2–1.3. The improvement for the instantaneous current estimates is similar. The mean and standard deviation values of the differences between observed and estimated currents for the delayed mode algorithm are summarised in Table 1, bottom panel.

## 5 Virtual AIS

Developed in the 1990's, the automatic identification system (AIS), which is based on VHF radio communications, allows ships to both see and be seen by other marine traffic in their area. The system augments radar and has increased the safety at sea. Since AIS instrumentation is generally bulky and would take a substantial cut from the glider's energy resources, and the fact that AIS signals do not penetrate water, it is for technical reasons not feasible to equip a glider with an AIS transmitter. Being able to broadcast its position to surrounding ships would, however, reduce the probability of a collision between a glider and a ship drastically. An alternative to AIS is virtual AIS, whereby the position of an object (glider) is broadcasted from an AIS transmitter elsewhere (a land station). In Germany the authority Wasser-und-Schifffahrtsverbund (WSV) regulates the use of virtual AIS, and has shown interest in this approach.

The principle of operation of a virtual AIS system is as follows. Two situations are discerned, namely the period when the glider is at the surface, and when it is underwater. When the glider is at the surface, and has established a (satellite) communication link with a server on shore, its actual GPS position is known. This information is instantly and automatically relayed to an operator room of WSV, from which the positional information of the glider is broadcasted as an AIS message. When the glider is underwater, and no actual GPS position is available, an estimated position can be broadcasted. To estimate a position, information is required on the local current field (drift), and the behaviour of the glider in terms of hardware behaviour (how it is programmed and consequently, how it reacts to the environment), and the dynamic behaviour (how and how fast it flies through the water). The modelling of the glider behaviour is considered beyond the scope of this study and therefore not discussed.

Assuming that an adequate model of the glider behaviour is available, it is furthermore required to quantify the drift due to the current whilst the glider is underwater. The drift can be estimated from integrating the estimated instantaneous (depth-averaged) currents over the period of the dive. The instantaneous current is computed as outlined in the previous section, except for some modifications. Since no new information can be taken into account until the glider resurfaces again, the residual current component cannot be computed from a linear interpolation during the dive. Instead, the residual current component is taken equal to its estimate at the time of diving, and held constant during the dive. For the same reason, using (16) and (17), the tidal current component is computed from the *a postiori* estimate of the state vector at the time of diving only[1].

It is expected that the uncertainty in the underwater glider position grows the longer it is underwater. The synthetic data set can be used to quantify the effect of subsurface time on the uncertainty in position, as this dataset can easily be divided in predefined subsurface times. Running the (forward) low pass and Kalman filters repeatedly for subsurface times, spanning 12 hours with 10 minute intervals, ensembles of six consecutive runs are formed. Figure 8 shows the ensemble averaged errors in estimated position for the mean, the 75 and 95 percentile errors, drawn by blue, green and red solid curves, respectively. The identically shaded areas indicate the variation present in each ensemble. As anticipated the errors increase with increasing

---

[1]It is, in fact, possible to use interpolated amplitudes, based on the *a postiori* estimates at the time of diving and the *a priori* estimates at the time of resurfacing, however, this brings no benefit because of (15), see also the conclusion of Section 4.3.

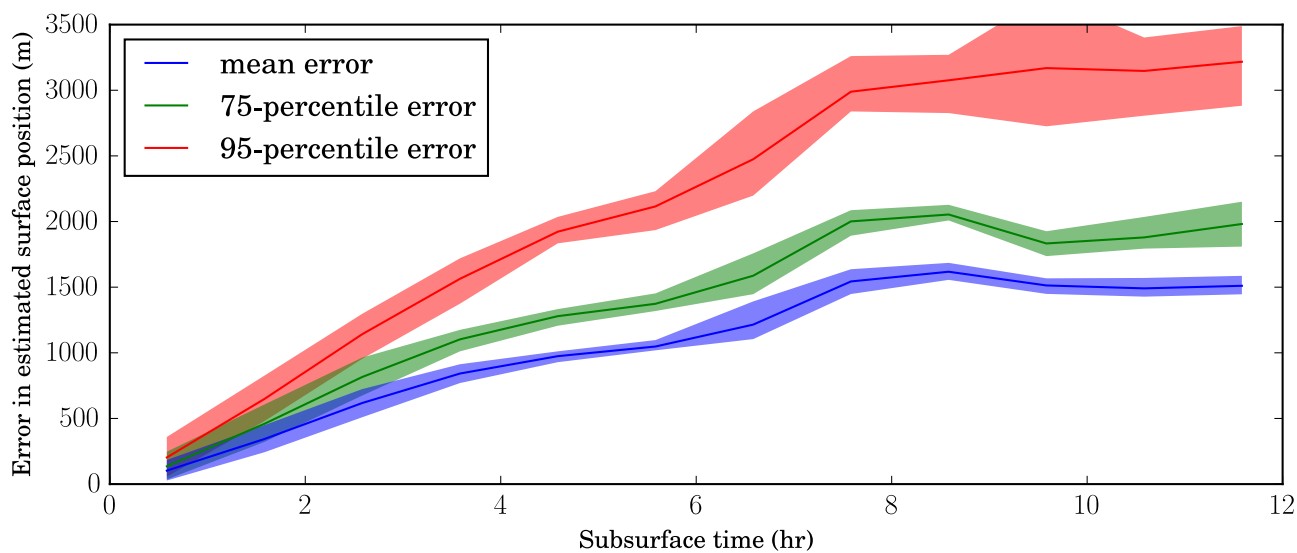

**Figure 8.** Error in estimated resurfacing position as a function of subsurface time.

subsurface time. Because of the longer integration times, the errors in the estimated velocities in fact reduce slightly with increasing subsurface time, expressed by the flattening of the curves for longer subsurface times.

As the position error increases from zero at the time of diving, the errors shown in Figure 8 are the maximum errors, *i.e.* the expected errors just prior to resurfacing. Depending on the required limit of this error, the maximum allowable subsurface
time can be defined. Here the data suggest that for a subsurface time of 3 hours, the average error is less than 700 m and that virtually all estimates are within double that distance.

Presently, this system is not implemented yet. The authority WSV has expressed its interest, and also indicated that the errors in the prediction for 3 hourly dives are acceptable. Technical limitations of the AIS system in use by WSV prevents a (semi-)automatic implementation. Furthermore, the range of the land stations to broadcast the AIS messages is limited to about
70 km offshore, and would not reach far enough to cover the outerparts of the German sector of the German Bight.

## 6   Discussion

The approach presented herein comes with a number of advantages. First, with a focus on glider path prediction, previous experience has shown that an unjustifiable amount of effort is required to guarantee current model output to be available at all times. Using glider estimated currents removes this vulnerability, as this information is always available, assuming a glider
operates normally. Second, the proposed algorithm provides independent estimates of the instantaneous currents. In near-real

time these estimated currents can be assimilated into the COSYNA-run ocean current models of the German Bight, in a similar fashion as radar observations of surface currents are assimilated (Stanev et al., 2015).

In delayed mode, when all data are available, the accuracy of the current estimates can be further improved. Still, the accuracy would remain inferior to the accuracy that can be achieved with direct measurements from devices such as ADCPs.

However, as often, for practical and logistical reasons, few, if any at all, independent current data are available that co-locate with glider data, so that a third advantage is that for many applications the improved glider based current estimates may be the only information on instantaneous currents available. This can facilitate the data analysis in studies involving gliders in tidal waters, similar to those published on phytoplankton blooms (e.g., Xu et al., 2013), sediment resuspension events (e.g., Glenn et al., 2008), or oxygen depletion events (e.g., Queste et al., 2016).

We chose to decompose the currents into a tidally driven part and a residual current. Lacking a realistic model for the residual currents, this component was quantified by a simple low pass filter, whereas the tidally driven currents were estimated using a Kalman filter based on the shallow water equations. Instead of this approach, a variety of other formulations could have been considered.

Instead of using a low pass filter, a Kalman filter for the residual current could be formulated based on the model

$$\dot{\boldsymbol{u}}_r = 0 \tag{24}$$

where the subscript "r" refers to residual. This model states that the current is constant. The Kalman filter will update the prediction of the current with every new measurement. How much the measurements are trusted over residual current modelled as a constant, depends on the predefined model noise. Similar to the forward low pass filter, this Kalman filter introduces a lag, the magnitude of which depends on the model uncertainty. As the purpose of this Kalman filter is to filter out the residual

component of the current, it is not straight forward to set the process noise such that right model stiffness is achieved. This is in contrast with the low pass filter, the behaviour of which is well-defined given the order and the cut-off frequency. If the model would be based on the assumption that $\dot{\boldsymbol{a}}_r = 0$, the same arguments still would apply, although this variant has the advantage that between measurements the estimated residual current is assumed to vary linearly, rather than being constant.

A different approach could be to incorporate the residual current Kalman filter based on (24) directly in the Kalman filter

developed in Section 3.2, by modifying (15) – (18). The major drawback of this formulation is that the filter has no means to discriminate between the residual and tidal components of the measured currents, other than specifying different model noises for the tidal and residual current models and may therefore not converge.

The robust solution, therefore, is to apply a low pass filter to the measured currents to separate the residual and tidal current components. Butterworth low pass filters of various orders and cut-off frequencies were considered. A first order filter with a

30 cut-off frequency of 1/24 cph was chosen *a priori*. This filter introduces a small group delay at the expense of attenuating only part of the main tidal signal. This raises the question whether or not a filter with other settings could fare better. Each of the filters discussed in Section 3.1 (see also Figure 1) was applied to compute the time and depth averaged currents in near-real time mode from the synthetic data.

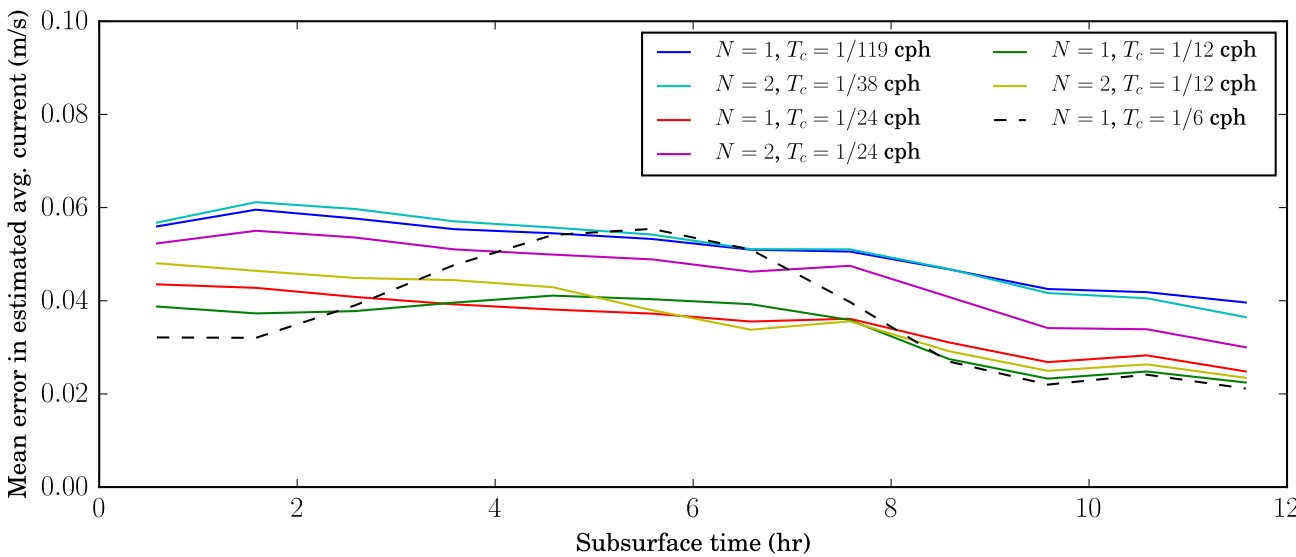

**Figure 9.** Mean error in estimated resurfacing position as a function of subsurface time for various Butterworth low pass filters with order $N$ and cut-off frequency $1/T_c$ cph.

Figure 9 shows the mean absolute error in the velocity estimate as function of the subsurface time for various filter settings. For the sake of convenience, the filter setting is denoted by a tuple consisting of the order and cut-off frequency in cph. The figure shows that if the filter design is based on removing almost all of the tidal signal, that is filter settings $(1, 1/119)$ and $(2, 1/38)$, both filters perform equally well. An improved overall performance is achieved, however, if the cut-off frequency is increased. This increase reduces the group delay introduced by the filter, but it also decreases the effectiveness of damping the main tidal signal. Although the filter settings $(1, 1/12)$ and $(2, 1/12)$ would also be acceptable, the filter setting $(1, 1/24)$ is considered performing best and has been used throughout this work. Increasing the frequency further, however, degrades the results, as is shown by the filter with setting $(1, 1/6)$.

## 7 Conclusions

Although the navigational algorithm implemented on board the (Slocum) glider yields depth and time averaged currents, the time resolution, set by the subsurface time, is often too coarse for the purpose of data analysis. This is particularly the case in regions where the currents are dominated by the tides. In this work an algorithm, tailored to coastal seas with strong tidal currents, was presented that can be used to estimate the instantaneous currents from the time averaged current measurements obtained by the glider. The algorithm considers a current component driven by the tides, and a residual current.

During a glider mission, the algorithm can be used to predict the currents, which is essential to make a projection of the glider trajectory up to 12 hours or so ahead. Run as a predictive tool, both the low pass filter and the Kalman filter are run forward in time only, which inevitably leads to lagging effects. This can particularly be apparent in the residual currents resulting from the low pass filter. For a typical application of a glider run in the North Sea, a first order Butterworth filter with a cut-off frequency of 1/24 cph was selected, being a trade-off between the amount of damping of the main tidal signal and the group delay.

To assess its performance, the algorithm was first applied to depth and time averaged currents, constructed from instantaneous currents measured with an ADCP, with known noise levels added (synthetic data). By averaging over time, information is lost, so that the measurements presented to the Kalman filter contain less information than the original ADCP measurements. The loss of this information is mostly compensated by information provided by the shallow water model. For an anticipated subsurface time of 3 hours, the correlation coefficients calculated for the estimated and ADCP measured instantaneous currents were found to be 0.97 and 0.96 for the eastward and northward components, respectively. This result indicates that the algorithm as such lives up to the expectations and is capable of reconstructing the instantaneous currents to a large extent.

When applied to the glider derived current measurement, and compared with ADCP data measured within a radius of about 10 km, the algorithm performs slightly worse, with correlation coefficients of 0.96 and 0.93 for the eastward and northward current components, respectively. This regression is attributed to the additional uncertainty caused by the navigation algorithm. Still, for subsurface times of three hours, which is a typical operational setting, the estimate of the instantaneous current has standard deviation of 6-7 $\mathrm{cm \cdot s^{-1}}$, which is considered low enough to be used for data assimilation procedures.

A further application could be to incorporate the present algorithm in a virtual AIS system to enhance the safety at sea. Herein the glider's position between surfacings can be estimated. The uncertainty in the estimated position grows with the time that the glider is underwater. Quantifying the effect of the uncertainty in the currents on the positional accuracy, it was found that subsurface times up to 3 hours would yield a positional accuracy that was still acceptable for the German authority Wasser-und Schifffahrtsverbund.

In delayed mode, the performance of the algorithm can be increased by running the low pass filter and the Kalman filter in forward-backward mode. The backward run in effect counters the lag introduced in the forward sweep. The standard deviation of the instantaneous current estimate was found to drop below 6 $\mathrm{cm \cdot s^{-1}}$. This means that for the purpose of data analysis, where the (depth-averaged) current is often regarded as an important driving force, the proposed algorithm provides a way to reconstruct the instantaneous currents with a sufficiently degree of accuracy.

*Acknowledgements.* This work was jointly financially supported by GROOM of the 7th Framework Programme of the European Union under Grant Agreement No 284321, and through the Coastal Observing System for Northern and Arctic Seas (COSYNA). The comments and suggestions of four anonymous reviewers are greatly appreciated.

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
