# Peer review of "Depth-averaged instantaneous currents in a tidally dominated shelf sea from glider observations"

_Biogeosciences, 2016_

## Referee Comment (RC1) · Anonymous Referee #3 · 13 Oct 2016

This manuscript compares a method to estimate depth- and time-averaged currents using a glider dead-reckoning algorithm with observed current data from an ADCP moored within 10 km of the glider's path. The ADCP 10-min data is averaged every three hours, low-passed to remove the tides and then added to tidal predictions using a Kalman filter, and finally compared with the glider data. The prediction errors are then estimated.

The objective of this work is worthwhile as the estimated barotropic currents can be used not only in conjuncture with the data from the other sensors on the glider but could also be assimilated in numerical models and, of course, be used to estimate the underwater position of the glider for navigation safety purposes. The only surprise is that the manuscript is submitted to Biogeosciences: a more suitable journal would be J. Atmos. Oceanic Technology, for instance. However, I understand that it has been

submitted to a BG special issue dedicated to the COSYNA program. The Introduction discussed possible applications and highlights the possible benefits of using his method to interpret the other glider data. A possibility is fine, but are there studies using this type of data being published in this special issue? Elsewhere?

As I mention, this is a technical paper and it should be published. The text is clear but it could be sometimes difficult to understand for somebody not in the field. The figures are also clear, but small. Moreover, I have a few questions and comments.

- Page 4, line 9; I assumed that a harmonic analysis has been conducted on the ADCP data: how much of the tidal variance is being accounted by the M2 tides? Probably a great deal if we look at figure 3, but it would be nice to know.

- Page 4, line 16; adding more tidal frequencies might not be as trivial as the phase information needs to be included. I am also surprised that a linear model (eq. 9) works so well in a shallow (40 m) region. However, the non-linearities observed on Fig. 6 do look small.

- Page 8, line 23; there is also an error associated with the ADCP data. For a 600 kHz, the error associated with 10 min (32 pings) averages of 40 cm bins would be close to 2.5 cm s-1. Averaging again over 3 h would reduce the random error under the bias error. I haven't checked lately, but RD-Instruments was suggesting that we should consider a bias error of a minimum of 1 cm s-1. I am mentioning this because I think that the author is underestimating the various errors associated with his method (as in Table 1).

- Page 8, line 29; I doubt that the buoyancy currents scale, the internal Rossby deformation radius, would be smaller than 8 or 10 km (I don't have the data to compute it). This means the glider would have to be within 3-4 km of the mooring. I think the results presented here simply mean that the tide, as mentioned in the manuscript, is the dominant feature.

[Figure]

- Page 9, line 7; see my previous comment.

- Page 11, line 1; . . . "the errors seem Gaussian". Has that been tested?

- Page 16, first paragraph; I am not convinced that this method will allow to incorporate the aspects of mixing in data analysis. I don't see how, except maybe in a Simpson and Hunter (1974) frame: tidal and wind mixing by friction.

In summary, the manuscript is publishable as it introduces a simple and clever method to estimate the barotropic, time- and depth-averaged currents, using the glider positioning algorithm. I agree with the authors: it is most useful in tidally-dominated regions. In a highly stratified region, one would need to get closer than 10 km from a mooring to assess the influence of stratification. Finally, I don't understand how the author can incorporated mixing with this method.

---

## Author Comment (AC1) · 7 Nov 2016

Response to Interactive comment on "Depth-averaged instantaneous currents in a tidally dominated shelf sea from glider observations" by Anonmous Referee #3.

I thank the Referee for his time and sharing his thoughts and comments on the manuscript "Depth-averaged instantaneous currents in a tidally dominated shelf sea from glider observations". I agree with the Referee that at face value the choice of the journal seems ill-chosen. As also noted by the Referee, the manuscript, however, is meant to be part of a special issue on Cosyna, a coastal observatory in the framework of which this work was carried out.

The Referee asks wether there are any studies published that might use this type of data (instantaneous currents observed from gliders), as the introduction only points out

the possibility of using this type of data. In fact, the incentive for this study was two-fold. First, the operation of gliders in the German Bight is controlled by the German shipping authorities, who require 12-hour position forecasts during glider operations. The methodology described herein has been used in this context for recent glider deployments. Second, in the German bight sediment resuspension events are mainly driven by (tidal) currents and wind waves. Current and yet unpublished research looks at resuspension rates and their relation with the tidal currents. I made a small change to the discussion (now page 16, line 31) where I come back to the issue for which studies this method could be applicable. Here I name a few studies that could use this type of data, if the would have been done in a tidal setting.

Specific comments:

page 4, line 9. Indeed a great deal of the tidal variance is accounted for by the M2 tide. Although not so essential in this context, I supplied the numbers 80% and 65% for the eastward and northward currents, respectively.

Page 4. line 16 (I think meant is page 6). Technically adding more tidal frequencies is trivial. The phase information is found by the Kalman filter (by means of the amplitudes a cosine and sine component). The linear model (eq 9) serves as a model to be used in the Kalman filter. The nonlinear terms ommited, notably friction, would, when included in the modelling, lead to interaction of signals with different frequencies and generate higher harmonics. The time scale at which this evolves is deemed longer than a tidal cycle. For the predictive capability of the method, adding nonlinear terms unnecessisarily increases the complexibility. For the reconstruction of the instantaneous currents, the Kalman filter is set flexible enough to allow for the tidal components to change slowly over time (through the modelling uncertainty). Since gliders can move (slowly) in space, it is to be expected that tidal components can change slowly in time, so that also in this case the added complexity of nonlinear terms, leaing to a nonlinear Kalman filter, brings no real benefits. Good point, though.

Page 8, line 23: It is true that there is also an error associated with the ADCP readings. Also that these errors might not have a zero mean. In order to test the algorithm per se, I make use of syntethetic currents, derived directly from the ADCP data in order to be realistic. Before adding the Gaussion noise the synthetic data are assumed to be the true observations. What the glider observes is computed from the synthetic data. The reason for this is that then the effect of different subsurface times can be estimated, but also the uncertainty in the ADCP measurements can be ignored. See also the first paragraph of section 4.2.

Page 8, line 29. I totally agree with the Referee. What was meant here is that buoancy driven currents on large scales, because of the fresh water influx of the River Elbe for example, would have a non-tidal, but barotropic character (far enough away from the river mouth). To remove any ground for confusion I changed the text to refer to "tidal currents and mesoscale circulation"

page 11, line 1. "The error seem Gaussion". This has been tested using a standard chi-squared based test, which does make me believe the hypothesis that the errors are Gaussion, using a level of significance of 5%.

page 16. I believe I have confused the author to how the calculated instantaneous currents could contribute to the analysis of glider data in terms of mixing. I have rephrased that in the discussion so that it should be clear as what is meant. (second paragraph of section 7)

---

## Referee Comment (RC2) · Anonymous Referee #4 · 14 Nov 2016

The manuscript, "Depth-averaged instantaneous currents in a tidally dominated shelf sea from glider observations" proposes a novel and useful method to address the challenge of estimating the local currents being experienced by a glider. This is important not only to glider safety (i.e. the example of posting warnings to mariners given in the manuscript), but also more generally to glider navigation and improves a key scientific observation from gliders, as currents are important to the characterization of marine systems. The method is rigorous and, although I do not know the literature thoroughly, appears to appropriately reference related work. In general, the presentation is good, though in the comments that follow I suggest a number of changes that would improve the clarity and in one case the scope and therefore potential readership of the manuscript. I suggest that it be published after the minor corrections listed below are addressed.

I have two general comments:

1. Given the drawback of the 10 hour phase lag in the near real-time current estimates, as I was reading I kept asking myself why you wouldn't have chosen a filter with less delay. You do discuss/show three different Butterworth filters and discuss their relative merits, but the discussion seems too narrow (no discussion of non-Butterworth filters) and too abstract (no sensitivity analysis of the consequences of using a filter that has a less sharp division between pass and not-pass but less delay). I would suggest including a fuller discussion of the relative benefits/disadvantages of using a less accurate but lower delay filter in the near realtime application – also considering the possibility of 1)using different filters for realtime and post-processing, or 2) if the delay is known, shifting the data appropriately.

2. I think this manuscript stands to increase its significance and gain much wider readership if it included a fuller discussion of how to generalize and apply this method in regions with different current regimes (e.g. mixed tidal components or non-tidal coastal currents). Although the focus on COSYNA may be appropriate to the special issue and is certainly an excellent test case, the method could be useful to the wider glider user community, but readers are more likely to pick it up if the manuscript provides help in understanding how it might apply to their region of operations.

Specific Comments:

Page 3, Lines 26-27: Why mention, and then leave out, the short non-tidal timescales? Are they not important to the predictions needed for the GVTCC? Or, is it simply too difficult? Please address this or remove the mention.

Page 4, Lines 2-4: Please rephrase this. It's strangely worded and unclear as to whether this refers to all filters or your choice of filter.

Page 4, Lines 13-14: Please give a concrete example to support the sentence "In practise this would mean that the algorithm would account for a transient feature, but

with a given time delay yielding poor results throughout the time span of the delay."

Page 5, Eqns 2-6: These equations need to be better explained. There are undefined terms and it is unclear how they are applied.

Page 6, Eqns 7-8: How are these estimated?

Page 8, Lines 11-18: Paragraph is awkward and as written does little to help the reader understand what you did (for example I didn't understand that the first test was based on purely synthetic data until I read on). Please reword for clarity.

Page 8, Lines 28-29: What argument is this expectation based on? Can you add a scaling argument to convince the reader?

Page 9, Lines 6-7: Please provide the motivation for the choice of noise being "Gaussian with zero mean and a standard deviation of 1 cm Âůs $-1$"

Page 10, Fig 3: It looks like more than just the phase changes between the forward-only and forward-backward filters. Tidal frequencies are still present in the forward-only. Please discuss this.

Page 11, Line 4: How large a problem is the error margin of 9-9.7 cm/s? Please provide some context.

Page 11, Lines 13-14: Please explain to logic behind this statement "The sources of these errors are the decomposition of the currents into residual and tidal currents by low pass filtering on the one hand, and the oversimplified tidal currents model on the other"

Page 16, Lines 1-3: Confusing as written. Please explain your reasoning. Why leap from currents to mixing? Why do you need currents to assess mixing? Are not currently intrinsically interesting or necessary to other applications? (e.g. dispersal of pollutants, larvae, etc).

Technical comments:

Page 5, Line 3: Meaning of "odd-padded"?

Page 6, Line 10: Change "casted into" to "cast as"

Page 8, Line 25: Change "tracks are" to "track is"

Page 9, Line 2: Change "In order to evaluate the performance of the filter, first the ADCP measurements were used." to "As a first step, the ADCP measurements were used to evaluate the performance of the filter."

Page 14, Line 18: Unclear. What is meant by "and responses" in this sentence?

---

## Author Comment (AC2) · 25 Nov 2016

Response to Interactive comment on "Depth-averaged instantaneous currents in a tidally dominated shelf sea from glider observations" by Anonmous Referee #4.

I very much appreciate the comments and suggestions from the Referee. Two general comments were given. The first comment suggests a fuller discussion on the filter design. I have chosen to keep the focus on Butterworth filters only, mainly because they are fairly simple and used ubiquitously. Discussing elliptic or more exotic formulations would bring no benefit to sorting the problem of eliminating the non-tidal current signal from the measurements. The question why not using a lower order filter that has also a lower group delay is a valid one that I took at heart. In fact, I looked at it in the early stages of the project, and concluded that the reasoning given in the manuscript lead

to the better filter of the ones discussed. Until I looked at the issue again and found a programming error that, when fixed, to my shock, changed the results significantly. For the better, fortunately. Therefore, I have expanded the section on the filters, to include other settings, and chose a first order filter with a cut-off freuquency of 1/24 cph. This filter produces significantly better results compared to the one I used in the previous version. I have updated the new results throughout the manuscript, and felt that it might be convenient to summarise those in a table for a quick overview. I have devoted another paragraph and graph to the discussion, where I address the question why I chose this particular filter setting and showed that other settings produce inferior and some similar results.

Sections modified: 3.1 inclusion of different filters, chosing a different filter than in previous manuscript 4.2 updated the numerical results/graphs 5 idem 6 added paragraph on the influence of the filter setting.

The second general comment is meant to add a discussion of a generalisation of the approach. This, to extent the potential readership. I did not follow this up and I will try to explain why. The problem at hand is that due to the way the glider navigates, time averaged current estimates are available. Because of the averaging, information on the variability during the dive is lost. In case of a tidal sea, as in this work, the tidal motion is responsible for a big part of the variability of the currents during the dive. And since the tidal motion lets itself model well, a Kalman filter can be constructed from this model to recover most of the information lost. As such, this idea is very general: use a model/Kalman filter to improve the knowledge of the state of a system. However, to broaden the scope of the paper and discuss how this can be applied to a situation that is non-tidal but has a significant variability on time scales that are less than the subsurface times, and is easy to model, seemed not so straight-forward. I, at least, could not find a convincing example. Perhaps when a glider would be deployed in a lake where seiches occur.

Specific comments
P.3 line 26-27 : I removed the mention as suggested. P4. 2-4, 13-14: rephrased as requested. P5. eqs 2-6. : I have included an explanation of the stepwise execution of the filter. That is, each time a measurement becomes available, the eqs 2-6 are computed. It is explained for each step what it does and why, including the estimation of the initial conditions. (next point) P6 eqs 7,8 see previous item. P8. lines 11-18: reworded P8 28-29: The same point was raised by referee #3, and already addressed. P9, 6-7. The motivation was already implicit in the manuscript. A clarification is added to make that explicit. Previously, there is a small discussion on how accurate a depth-time-averaged current measurement can be, depending on the errors of GPS and the subsurface time, and navigation model errors. The estimate is some 1 cm/s. See also equ 19.

p10 line 3. The forward filter does not remove all the tidal signal, and causes a delay. This is now mentioned explicitly. Also the stress on the delay is reduced a bit, because of the different filter setting used now, the delay is not that significant anymore. Still visible though. P11, line 4. Thee error margin is put into context to how much the position can be forcasted for a 3 hour dive, also the topic of the section on virtual ais. P11, l13-14. I left this section out, as it is confusing. P16, lines 1-3. This also confused referee#3 and was already addressed. Basically, no connection to mixing is made anymore.

Technical comments:

The use of odd-padded is removed. It is basically an option in a filter call, that has only marginal effect if any. Leaving out, or using a different option would not give any different results or conclusions.

Other technical comments have been applied, and last techincal commetn P14, l18 has been clarified. With responses is meant how the glider reacts to the environment. Reworded accordingly.

The comments of this reviewer have given me quite a bit of extra work, but I am very

thankful for that, as it greatly improved the results of this manuscript.

Lucas Merckelbach